

# Electrical resistivity imaging data for hydrological and soil investigations of virgin Rospuda river peatland (North-East Poland).

Grzegorz Sinicyn[1], Radosław Mieszkowski[2], Łukasz Kaczmarek[1], Stanisław Mieszkowski[2], Bartosz Bednarz[1], Krzysztof Kochanek[1], Mateusz Grygoruk[3], Maria Grodzka-Łukaszewska[1]

[1]Faculty of Building Services, Hydro and Environmental Engineering, Warsaw University of Technology, Warsaw, 00-653, Poland
[2]Faculty of Geology, University of Warsaw, Warsaw, 02-089, Poland
[3]Institute of Environmental Engineering, Warsaw University of Life Sciences, Warsaw, 02-776, Poland

*Correspondence to*: Łukasz Kaczmarek (lukasz.kaczmarek@pw.edu.pl)

**Abstract.** This publication presents data on geophysical measurements performed in the Rospuda wetlands located in North-Eastern Poland. The measurements were carried out by means of the the Electrical Resistivity Imaging (ERI) method, which so far was to our best knowledge never used in the River Rospuda wetland valley. The ERI data were collected in single survey campaign in November 2022 to account for the wet season. During the campaign two ERI profiles were measured. The aim of the field works was to provide the material for illustration of the arrangement of geological layers creating the wetland. The

data repository contains detailed data descriptions for each survey site. The ERI data from the selected survey sites can be used first of all to create the conceptual numerical model of groundwater and surface water interaction in this environmentally valuable area, which is to a certain extent a scientific terra incognita, but also for hydrological investigation of hydraulic conductivity and hydrodynamic field, identify geological structure, and characterize engineering properties of the organic soils.

## 1 Background

Peatlands constitute unique areas of the interaction between soil, groundwater and surface water. The organic soils, extremely valuable for the environmental, due to their accumulation properties (water, carbon dioxide, organic matter), high compressibility and variable water permeability depending on the tension level, constitute a great challenge in planning their protection and potential development. Due to challenging availability and high variability of geological conditions, all the data considering wetlands become a very important input for further analyses, numerical calculations, and field and laboratory tests.

Mostly because of the environmental uniqueness of this case study areas, the non-invasive method of electrical resistivity tomography (which does not need heavy transportation to be moved), fits perfectly into the circumstances of achievable tests (i.e. Kowalczyk et al., 2017). The research is of reference nature for peat bogs occurring in this part of Europe, because of the scope of research and the research techniques used which additionally allowed for continuous identification of soil variability in the subsoil area. The peatland developed in the large land depression in Holocene (deglaciation of a Würm-Vistuilan Phase)

that forms Rospuda River Valley. This area is representative for many similar peatlands in northeastern Poland. Rospuda River





Valley hosts one of the last large, well-preserved percolation mires (rich fen system) remaining in Central Europe (Jabłońska et al, 2010; Jabłońska et al., 2011; Jabłońska et al., 2014; Jabłońska et al., 2019). Figure 1 shows the geological structure (of the locations where the geophysical prospecting was done) in the form of cross-sections (Jabłońska et al., 2010) made on the basis of reconnaissance drilling.

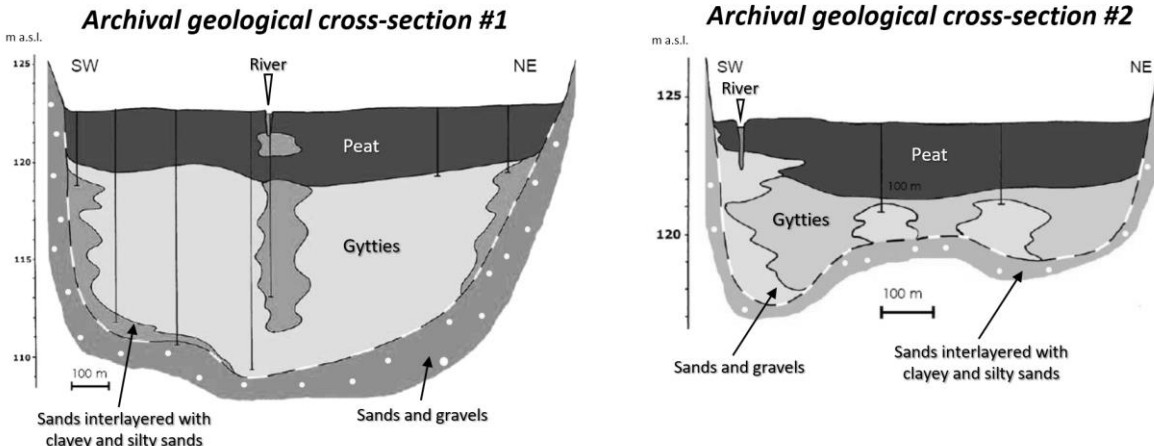


**Figure 1: Archival geological cross-sections in the Rospuda research site (modified Jabłońska et al., 2010).**

## 2 Data description

The data consider electrical resistivity imaging results for hydrological and geological investigations conducted in the reference
Polish location (Figure 2). They can be easily accessible at Mendeley's data repository (Sinicyn et al., 2024): https://data.mendeley.com/datasets/10.17632/5m34cs5zn4/1. The repository is structured into a folder - *ERI Rospuda Data*, which contains: raw data (presented general array format; in "dat" files) and inverted resistivity models images for each ERI profile (presented in "jpg" files). The Google Earth KML (Keyhole Markup Language) files with the location of the ERI profiles are also provided for each survey site. The data are of comparable high quality because all the ERI profiles have been
measured by the same ABEM Terrameter LS-2 setup by Multiple gradient array (Roll Along technique) with 5 m (profile #1) and 2 m (profile #2) electrode spacing.

Below, Table 1, field photos, reference resistivity table, location map and selected exemplary results can be found. Table 1 presents details on the ERI's metadata: profile line name, survey data, beginning and ending coordinates points of ERI profiles, elevation, profile orientation, array type, electrode spacing, profile length, filer type, file name, instrument info.

**Table** **1**

ERI data resume descriptions for the Rospuda site.





| Profile line name | Survey date (dd/mm/year) | ERI profile coordinates (X;Y in EPSG 2180) | | Min and max elevation (m a.s.l. in PL-KRON86-NH) | Profile orientation | Profile length (m) | File name (and type) |
|---|---|---|---|---|---|---|---|
| | | Begin | End | | | | |
| 1 | 26/11/2022 | 678040.9; 759131.6 | 678466.9; 759880.8; | 122.5; 131.5 | SW->NE | ~900 | ERI profile 1 (dat) |
| 2 | 26/11/2022 | 680204.3; 757863.8 | 680452.3; 757997.5 | 124.4; 124.8 | NW->SE | ~280 | ERI profile 2 (dat) |







Figure 2: The ERI data distribution on the topographic map background (source: www.geoportal.gov.pl). Dark blue colour means the lowest elevation, where dark red the highest one).


## 3 Experimental design, materials and methods

The two-dimensional resistivity imaging data were collected by galvanically injecting a low-frequency electrical current into the ground via two electrodes and measuring the voltage difference between two potential electrodes (methodology based on Loke, 2018). Differences in resistivity values caused by the flow of electric current through various subsurface mediums are



used to identify materials (i.e. materials listed in Table 3). Electrical resistivity of the subsurface material is determined by the composition of soil (particle size distribution, mineralogy), its structure (porosity, pore size distribution, connectivity), fluid content, the concentration of dissolved electrolytes, the clay content, and the temperature [Palacky, 1988; Loke, 2018; Tarnawski, 2020]. Table 3 depicts the electrical resistivity/conductivity characteristics of common subsurface geological materials in this area of Poland.


**Table 3: Typical electrical properties of resistivity (inverse of conductivity) of common geological materials and other mediums in Poland (based on Tarnawski, 2020 and authors' experience).**

| Soil or other medium | Typical range of electroresistivity ($\Omega$m) |
|---|---|
| Clayey deposits (clay, till with clay) | <25 |
| Organic soil (peat, alluvion) | 10-100 (in aeration zone: 30-100; in saturation zone: 10-50) |
| Tills and loams | 25-70 |
| Sandy deposits | 70-1000 (in aeration zone: 200-1000; in saturation zone: 90-250) |
| Surface water | 0.1-300 |
| Rainwater | 30-1000 |
| Mineralised water (i.a. sea water) | 0.1-5 |
| Permafrost | high |

The advanced multi-electrode resistivity sensors were used to measure numerous data points in a single ERI profile by
automatic switching of the current and potential electrodes. A multiple-gradient array was used to collect resistivity data in the both forward and backward survey directions. Figure 3 depicts the acquisition of field ERI data with the one of the most advanced 12-point light ABEM Terrameter LS-2 setup (Figure 3a). The electrodes were hammered along the profiles and connected to cables with the cable joints for 21 take-out cables , which lead to a resistivity meter during resistivity measurements. The electrodes were tested for the contact resistance before each measurement session, and apparent resistivity
was measured. Then, apparent resistivity in many of data points can be measured for a single ERI profile (eventually, giving the effect of *quasi* continues section). The multiple-gradient array was described in detail i.a. by Loke (2018).







**Figure 3: Acquiring ERI field data with multi-electrode (a) 12-point light Abem Terrameter LS-2 set-up at ERI profile**
**No. 1,  (b) raw visualization of results; (c) ERI profile No. 2.**

The measured datasets were filtered to minimise noise. The RES2DINV software package (Aarhusgeosoftware Manual, 2022) was used for data processing and inversion. The smoothness-constrained least-square (DeGroot-Hedlin and Constable, 1990) and robust inversion (Wolke and Schwetlick, 1988) algorithms were used for data processing, depending on the expected
subsurface features, however, the ERI data were mostly inverted by means of the smoothness-constrained least-square inversion algorithm. The iterative inversion method was applied until the discrepancy between measured and predicted resistivity data reached acceptable levels, i.e. when the root mean square (RMS) error dropped below 5% (information about the difference between the measured and calculated apparent resistivity values). This value may  be exceeded for surveys in hard rock and noisy environments.  As an example, we show the ERI data distribution (Figure 4) for resistivity data collected
from the Rospuda site – prospection depth: ca. 32 m (profile #2) and 43 m (profile #1) b.t.s.



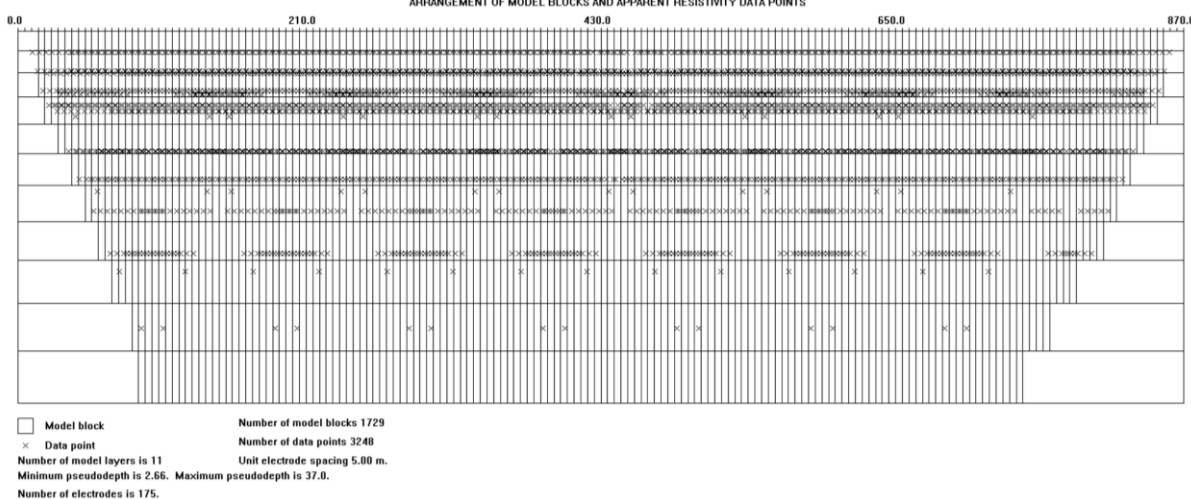

**Figure 4:  ERI profile No. 2 at the Rospuda site: the electrical resistivity data points distribution.**

Using the Roll Along technique, it was possible to obtain very valuable long prospecting lines (several hundred meters long).
Figure 5a shows the distributions of apparent resistivity data, Figure 5b shows model calculated apparent resistivity data, and
Figure 5c shows an inverted resistivity model.







Figure 5: ERI profile No. 1 at the Rospuda site: (a) apparent resistivity data, (b) calculated apparent resistivity data, (c) inverted resistivity model, and (d) inverted resistivity model with topography.

## 4 Limitations

While conducting the research campaign by the ERI profiling we faced difficulties related to the high water saturation impact on the current field; in such a case the Terrameter revealed false data with zero or negative resistivity values, or quite contrary, large resistivity variation (especially near the surface). Such unexpected obstacles lead to the time-consuming results processing or even made the measurements impossible. Fortunately during the investigation there was not rain and we manage to keep ERI set up not too much wet – thus we avoided the previously mentioned problems. Moreover, we managed to achieve





a very low level of absolute error in iterations - in the first long profile this error was just 1.3% after six iterations and in the second case it was just 0.94% after 5 iterations.

## 5 Data value

The Electrical Resistivity Imaging data from the selected survey site can be used to perform numerical modelling of groundwater and surface water interaction (i.e. van Loon et al., 2009; Grodzka-Łukaszewska et al., 2022) in the environmentally valuable area of the River Rospuda valley which is to a certain extend a scientific terra incognita, but also for hydrological investigations of hydraulic conductivity and hydrodynamic field, identify geological structure, and characterize engineering properties of the organic soils.

The ERI data can be used to monitor groundwater heads (Chang et al., 2023) or generally hydrogeological conditions (i.e. Kowalczyk et al., 2014) as well as terrain changes (mainly subsidence due to water loss in the wetland's body) by comparing with future surveys. It can be related to the climate change effects.

The ERI data can be jointly inverted and interpreted with other field measurements (i.a. other geophysical methods, boreholes) to obtain more reliable subsurface information. Studies such as recognition drilling and probing (i.e. Carrière and Chalikakis, 2021), sampling and hydraulic conductivity lab tests (Kaczmarek et al., 2023), low-flow filed pumping tests, seismic, electromagnetic, and ground penetrating radar can be effectively integrated with ERI data.

Through open-source inversion algorithms, raw ERI data can be reprocessed to generate 2D and 3D inverted models (i.e. Cockett et al., 2015). Machine learning and statistical algorithms can be used to further interpretation of the inverted resistivity data.

## Data availability

Datasets used in this article: Sinicyn, G., Mieszkowski, R., Kaczmarek, Ł., Mieszkowski, S., Bednarz , B., Kochanek, K., Grygoruk, M., Grodzka-Łukaszewska , M.: Electrical resistivity imaging data for hydrological and geological investigations of virgin Rospuda river peatland (North-East Poland) - original data, Mendeley Data, V2, doi: 10.17632/5m34cs5zn4.2, 2024.

## Author contributions

SG: Conceptualization, Methodology, Data Collection, Writing- Reviewing and Editing. MR: Methodology, Data Collection, Data Processing. KŁ: Conceptualization, Methodology, Data Collection, Data Processing, Validation, Writing, and Original draft preparation. MS: Data Collection, Data Processing. BB: Data Collection, Writing- Reviewing and Editing. KK: Data



Collection, Writing – Reviewing and Editing. MG: Writing – Reviewing and Editing, Funding Acquisition. G-ŁM: Conceptualization, Methodology, Data Collection, Writing, Original draft preparation.

**Competing interests**

The authors declare that they have no known competing financial interests or personal relationships that could
have appeared to influence the work reported in this paper.

**Acknowledgements**

The research leading to these results has received funding from the Norwegian Financial Mechanism 2014-2021 (project no. 2019/34/H/ST10/00711).

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
