# Peer review of "Electrical resistivity imaging data for hydrological and soil investigations of virgin Rospuda river peatland (North-East Poland)."

_Geoscientific Instrumentation, Methods and Data Systems, 2024_

## Referee Comment (RC1)

**Electrical resistivity imaging data for hydrological and soil**

**investigations of virgin Rospuda river peatland (North-East Poland).**

**Geoscientific Instrumentation Methods and Data Systems**
**gi-2024-11**

Overall:

The paper brings results of an interesting research on the electrical resistivity imaging data, specifically for soil investigations in North-East Poland.  However, there are few issues which should be clarified and improved before publishing the manuscript. Therefore, I recommend the paper for the publication after **minor revisions**.

Comments and suggestions:

1. An abstract should be less descriptive and should contain achievements of an the authors and results of their research.
2. The present state review should be added/extended in the Introduction. More international publications on the topic should be mentioned and discussed. ERI techniques should be mentioned and discussed in more detail.
3. The more detailed map should be included before the cross sections are depicted. The profiles should be marked in the map. I recommend to move map and profiles to the section dealing with the site, not to be in the Introduction.
4. Fig.2 As mentioned above the more readable map should be included first (overall tiny map in the upper left corner is not enough. After that detailed map with marked profiles should be included, the profiles should follow such map e.g. as Fig. 3.
5. Fig. 2 It should be clearly explained what is depicted in the Fig. 2, meaning of colours (legend) should be included and comments and interpretation of the colours should be attached in the text.
6. L72 - "driven" instead of "hammered"?
7. Fig. 4 is hardly readable. Again it is not very clear what is depicted and how to interpret it. Please, complete.
8. Fig. 5 the scale along the axes and the legend is not readable. Please, improve.
9. L114 - How hydraulic conductivity can be derived from ERI, what is the accuracy of such estimate?
10. Conclusions are missing, please complete.
11. The literature review should be extended and related to the other scientific studies dealing with ERI.

---

## Author Response (AR1)

**CC1: 'Comment on gi-2024-11', Vincenzo Lapenna, 22 Nov 2024 reply**

The paper deals with the application of the method of Electrical Resistivity Tomography for the investigation of the near-subsurface in a complex hydrogeological area: the Rospuda river bog (Poland). To date, there has been a growing interest in novel applications of the ERT method, such as to study peatlands and the impact of climate change on these fragile geological environments. This paper is therefore welcomed.

However, I suggest to improve the overall quality of the paper.

1. In the Introduction section, it is important to cite the most recent and relevant papers on the subject.

**We have expanded the literature. Thank you for your advice.**

2. It is necessary to improve the quality of Figure 2, the small figure at the top is not clear. It is important to localise the study area in a large scale.

**Thank you for this prompt - we have added new localisation figure on a large scale.**

3. The order of figures 1 and 2 could be changed.

**We used this helpful suggestion.**

4. The paragraphs 2 and 3 could be combined. The data could be described after the presentation of the experimental field design and data collection.

**You are right – we have introduced such text modifications.**

5. The Data value and Limitations paragraphs could be merged into a "Discussion" paragraph.

**Thank you for this remark – we added a Discussion section.**

6. The analysis of the results is purely qualitative, I suggest to include more quantitative considerations and a comparison with the hydrogeological data.

**We have improved the paper by adding i.a.:**

The obtained electrical resistivity images revealed the geological structure and the position of the water table very well. On the electrical resistivity cross-sections it is possible to distinguish: (i) a zone of very high resistivity, which corresponds to sands and gravels with very low water content constituting the high ground (a few thousand  $\Omega m$ ; maximum ~4310  $\Omega m$ ); (ii) a zone of low resistivity (from ca. 75  $\Omega m$  to ca. 500  $\Omega m$ ), occurring below the near-surface zone and which are most probably fully water saturated sands; (iii) a zone of even lower resistivity than Zone II (from ca. 40  $\Omega m$  to approx. 75  $\Omega m$ ), which are water saturated silty sands (which may be overlain by clayey sands) - this zone is present at a relatively great depth of approx. 30 m; (iv) a zone of the lowest resistivity (10-40  $\Omega m$ ), which is water-saturated peats of a thickness ranging from a few to

several meters at the maximum (the highest thickness of peats is present near the Rospuda River). This interpretation corresponds to the geomorphology of the described area (detailed by Ber, 2007): the Rospuda River valley is a subglacial trough adjacent to sand plains in the west and east, within which numerous eskers and other forms of crevasses are present. This type of form was identified as an elevation in the first cross-section. The Rospuda River heads southeast, flowing approximately 2.5 km further into Lake Rospuda. Rospuda is a drainage river type, with groundwater recharge from the northwest and east. The groundwater table is in intense contact with the surface water of the Rospuda River. The influence of the water table level in the sand and gravel-type soils (Zone I) changes their electrical resistivity image (first cross-section), causing a difference in resistivity of about 1000  $\Omega$ m (a decrease from about 4000  $\Omega$ m to about 3000  $\Omega$ m). This description indicates a thick Quaternary groundwater aquifer (no impermeable soil to a depth of about 43 m) with no more significant surface isolation layers in the form of soils like clays or tills. In this context, peats constitute a separate hydrogeological layer characterized by lower hydraulic conductivity than the surrounding mineral soils but with excellent retention properties.

7. Finally, the English for needs to be revised.

Thank you for your feedback. We tried to upgrade the manuscript.

Many thanks for constructive participation in the discussion on our study.

**RC2:**

The paper can be useful for having ERT data in peatland scenarios. However, some changes are necessary to accept the work. Authors should enhance the resolution of the images and improve the introduction citing the most relevant scientific works regarding the analyzed target. Indeed the bibliography is too limited. As regards the data presented, it is interesting that the users provide information about data error measurement distribution. It is also useful for the reader to define for the adopted array used, the maximum reciprocal distance selected for potential and current electrodes

**Thank you for your collaboration. Based on your suggestion, we have improved our paper.**

Further, thanks to the availability of geological models a more effective interpretation of the inverted data should be provided. Are available direct data (i.e. cores) close to the ERTs?

**Unfortunately, there are no closer boreholes. We have done several, but they were shallow (1-2m) and they all were showing peats.**

The reference Jablonska et al. 2010 is difficult to use since it is not in English.

That is challenging. Mrs. Jabłońska published these two cross-section only in Polish. Nevertheless, she also published another (but as a scheme) geological cross-section in a paper in 2014. We have added information: "Jabłońska et al., 2014 – a schematic geological cross-section of the Rospuda valley along the riverbed described in the English)".

The authors should give more information about the geological context or insert another reference to clarify the geological context. For this reason, the case study presented in the actual form is not very usable for other readers.

**• Thank you for your comment. We improved the paper in this field. We added:**

... Rospuda River Valley was formed as the peatland, which developed in the large land depression in the Holocene (deglaciation of a Würm-Vistuilan Phase – Figure 2). The valley of the Rospuda River is vast (in the region of analysis, it ranges from around 0,5 km to 1,5 km), which is related to the glacial history of this region. Nowadays, the meandering river is in the middle (south part of the research area) and close to the west bank (north part of the research area) of the valley mentioned above, which has around 2,5 km between geophysical prospection lines. The analyzed area is surrounded by a large forest complex, where the entire region is a Rospuda Valley Protected Landscape Area (which partly constitutes also the Professor Aleksander Sokołowski Rospuda Valley Reserve). ...

...This interpretation corresponds to the geomorphology of the described area (detailed by Ber, 2007): the Rospuda River valley is a subglacial trough adjacent to sand plains in the west and east, within which numerous eskers and other forms of crevasses are present. This type of form was identified as an elevation in the first cross-section. The Rospuda River heads southeast, flowing approximately 2.5 km further into Lake Rospuda. Rospuda is a drainage river type, with groundwater recharge from the northwest and east. ...

Please, increase the quality of figure 4 (the electrical resistivity data points distribution). Why use a high vertical exaggeration when data distribution is so shallow?

**We wanted to highlight the depth prospection variability. Nevertheless, we reduced exaggeration and improved quality.**

Can you indicate the total number of rejected measurements for your inversion?

**The images show electrical resistivity cross-sections with no rejected measurements (no such critical need; several could be discussed, but we wanted to show the quality of "rough" data). We add to the manuscript this comment.**

Finally, there is no interpretation of the data. I know that the paper shows only the dataset but a short interpretation its fundamental for the correct use of the data.

**We are sorry for this inconvenience. We improved the paper in this field:**

The obtained electrical resistivity images revealed the geological structure and the position of the water table very well. On the electrical resistivity cross-sections it is possible to distinguish: (i) a zone of very high resistivity, which corresponds to sands and gravels with very low water content constituting the high ground (a few thousand  $\Omega$ m; maximum ~4310  $\Omega$ m); (ii) a zone of low resistivity (from ca. 75  $\Omega$ m to ca. 500  $\Omega$ m), occurring below the near-surface zone and which are most probably fully water saturated sands; (iii) a zone of even lower resistivity than Zone II (from ca. 40  $\Omega$ m to approx. 75  $\Omega$ m), which are water saturated silty sands (which may be overlain by clayey sands) - this zone is present at a relatively great depth of approx. 30 m; (iv) a zone of the lowest resistivity (10-40  $\Omega$ m), which is water-saturated peats of a thickness ranging from a few to several meters at the

maximum (the highest thickness of peats is present near the Rospuda River). This interpretation corresponds to the geomorphology of the described area (detailed by Ber, 2007): the Rospuda River valley is a subglacial trough adjacent to sand plains in the west and east, within which numerous eskers and other forms of crevasses are present. This type of form was identified as an elevation in the first cross-section. The Rospuda River heads southeast, flowing approximately 2.5 km further into Lake Rospuda. Rospuda is a drainage river type, with groundwater recharge from the northwest and east. The groundwater table is in intense contact with the surface water of the Rospuda River. The influence of the water table level in the sand and gravel-type soils (Zone I) changes their electrical resistivity image (first cross-section), causing a difference in resistivity of about 1000  $\Omega$ m (a decrease from about 4000  $\Omega$ m to about 3000  $\Omega$ m). This description indicates a thick Quaternary groundwater aquifer (no impermeable soil to a depth of about 43 m) with no more significant surface isolation layers in the form of soils like clays or tills. In this context, peats constitute a separate hydrogeological layer characterized by lower hydraulic conductivity than the surrounding mineral soils but with excellent retention properties.

**RC1**

**Overall:**

The paper brings results of an interesting research on the electrical resistivity imaging data, specifically for soil investigations in North-East Poland. However, there are few issues which should be clarified and improved before publishing the manuscript. Therefore, I recommend the paper for the publication after minor revisions.

**Comments and suggestions:**

1. An abstract should be less descriptive and should contain achievements of an the authors and results of their research.

**You are right – we have modified the abstract. Thank you.**

Abstract. The paper deals with the application of the geophysical method for the investigation of the near-subsurface fragile hydrological environments. Study delivers geophysical measurements data performed in the Rospuda wetlands located in North-Eastern Poland. The measurements were carried out by means of the the Electrical Resistivity Imaging (ERI; also called Electrical Resistivity Tomography, ERT) method, which so far was never used in this region of the River Rospuda peatland valley. The ERI data were collected in single survey campaign in November 2022 to account for the wet season. During the campaign two ERI profiles were measured. The aim of the field works was to provide the material for illustration of the arrangement of geological layers creating the wetland. The data repository contains detailed descriptions for each survey site. The study revealed a strong interaction between groundwater, characterized by a thick sandy aquifer, and surface water. In this system, surface present peat constitutes the dominant soil component within the contact zone between groundwater and surface water (with drainage river type). Variations in this relationship will have a direct impact on peat stability and associated hydrological processes. The water-saturated peat electrical resistivity zone (10-40  $\Omega$ m) have thickness ranging from a few to several meters at the maximum (the highest thickness of peats is present near the Rospuda River).

2. The present State review should be added/extended in the Introduction. More international publications on the topic should be mentioned and discussed. ERI techniques should be mentioned and discussed in more detail.

**Thank you for this suggestion. We have expanded the Introduction to include, among other things, the topic of electrical resistivity tomography, as well as doubling the number of references with international positions from the last several years.**

**1 Introduction**

Peatlands constitute unique areas of the interaction between soil, groundwater and surface water (Limpens et al., 2008). The organic soils, extremely valuable for the environmental, due to their accumulation properties (water, carbon dioxide, organic matter; Kane et al., 2019; Word et al., 2022), high compressibility and variable water permeability depending on the tension level (Paniagua, Long & L'Heureux, 2021; Long et al., 2022), constitute a great challenge in planning their protection and potential development. Due to challenging availability and high variability of geological conditions, all the data considering wetlands become a very important input for further analyses, numerical calculations, and field and laboratory tests. Mostly because of the environmental uniqueness of this case study areas, the non-invasive method of electrical resistivity imaging (ERI; which does not need heavy transportation to be moved), fits perfectly into the circumstances of achievable tests; i.e. Slater and Reeve, 2002; Kowalczyk et al., 2017; Minasny et al., 2019; Pezdir et al., 2021). ERI is a geophysical method used to investigate subsurface structures by measuring electrical resistivity variations. In environmental studies, it helps monitor soil water content distribution, soil structure (i.e., high resistivity diversity of peats: compare results of Pezdir et al., 2021 and Kaczmarek et al., 2024) as well as permafrost dynamics, which are important for assessing climate change impacts. Thus, it is particularly valuable for detecting changes in permafrost thawing, carbon-rich soil degradation (especially peats), and hydrological processes influenced by rising temperatures.

The research is of reference nature for peatlands occurring in this part of Europe, also because of the research technique, which used which provide continuous identification of ground variability. The study considers electrical resistivity imaging results of the exemplary section of the Rospuda River Valley, which structure is original (no anthropogenic influence). The field study is located in the East-Central Europe, North-Eastern Poland (close to the border with Belarus) – Figure 1.

3. The more detailed map should be included before the cross sections are depicted. The profiles should be marked in the map. I recommend to move map and profiles to the section dealing with the site, not to be in the Introduction.

**Thank you for the advice – we have improved the paper.**

4. Fig.2 As mentioned above the more readable map should be included first (overall tiny map in the upper left corner is not enough. After that detailed map with marked profiles should be included, the profiles should follow such map e.g. as Fig. 3.

**We agree with the suggestion – we have improved the paper (added maps with better scales).**

5. Fig. 2 It should be clearly explained what is depicted in the Fig. 2, meaning of colours (legend) should be included and comments and interpretation of the colours should be attached in the text.

**We clarified Figure 2 and added the necessary comments in the text above.**

6. L72 - "driven" instead of "hammered"?

Yes, we made the change.

7. Fig. 4 is hardly readable. Again it is not very clear what is depicted and how to interpret it. Please, complete.

**We improve this figure with adding information to the text about importance for providing a sketch of distribution of the acquired data.**

8. Fig. 5 the scale along the axes and the legend is not readable. Please, improve.

**Sorry for poor quality. We have improved Figure 5 (now Figure 7).**

9. L114 - Flow hydraulic conductivity can be derived from ERI, what is the accuracy of such estimate?

It has to be treated as estimation, but still quite powerful (it gives 2-D distribution). We added information about this with exemplary relation (in 4.2 Data value section)

It has to be treated as estimation, but still quite powerful (it gives 2-D distribution). We added information about this with exemplary relation (in 4.2 Data value section).

In case of hydraulic conductivity, it could be related to lab/field tests, which can be even use for further resistivity conversion into a hydraulic conductivity value (Coe et al., 2018 Kaczmarek, Dąbska i Popielski, 2024, although these values should be regarded as indicative). Determining a precise relationship between geophysical parameter values and the hydraulic conductivity is a complex challenge (Kirsch, 2009). Nevertheless, there are studies showing such (local) relation, i.e.  $R=8,14e^{1,23\cdot k}$ , where e is the exponential constant, R ( $\Omega$ m) is the average resistivity of the aquifer, and k (m/d) is the hydraulic conductivity of the aquifer.(Lu, Huang & Xu, 2021).

10. Conclusions are missing, please complete.

You are right; we added a summary section.

11. The literature review should be extended and related to the other scientific studies dealing with ERI.

**That's right. Sorry for the gaps in the article. We have improved it.**

**RC3**:**

This study focuses on applying Electrical Resistivity Tomography (ERT) to investigate the nearsubsurface of the Rospuda River bog in Poland, an area with complex hydrogeological conditions. With increasing interest in the novel applications of ERT, particularly in studying peatlands and their response to climate change, this research contributes to the growing body of work examining the effects of environmental changes on these fragile geological environments. The authors did a huge field work, but the paper need a strong revision. They have to work hard on in order to be a published paper.

Detailed suggestions:

**1 Background**

The background chapter is the final part of an Introduction chapter. I suggest to add this chapter ("Introduction") with dedicated details. It is important to describe the actual knowledges on the topic of the paper with the indication on relevant papers.

• We modified the Background section to Introduction form.

**1 Introduction**

Peatlands constitute unique areas of the interaction between soil, groundwater and surface water (Limpens et al., 2008). The organic soils, extremely valuable for the environmental, due to their accumulation properties (water, carbon dioxide, organic matter; Kane et al., 2019; Word et al., 2022), high compressibility and variable water permeability depending on the tension level (Paniagua, Long & L'Heureux, 2021; Long et al., 2022), constitute a great challenge in planning their protection and potential development. Due to challenging availability and high variability of geological conditions, all the data considering wetlands become a very important input for further analyses, numerical calculations, and field and laboratory tests. Mostly because of the environmental uniqueness of this case study areas, the non-invasive method of electrical resistivity imaging (ERI; which does not need heavy transportation to be moved), fits perfectly into the circumstances of achievable tests; i.e. Slater and Reeve, 2002; Kowalczyk et al., 2017; Minasny et al., 2019; Pezdir et al., 2021). ERI is a geophysical method used to investigate subsurface structures by measuring electrical resistivity variations. In environmental studies, it helps monitor soil water content distribution, soil structure (i.e., high resistivity diversity of peats: compare results of Pezdir et al., 2021 and Kaczmarek et al., 2024) as well as permafrost dynamics, which are important for assessing climate change impacts. Thus, it is particularly valuable for detecting changes in permafrost thawing, carbon-rich soil degradation (especially peats), and hydrological processes influenced by rising temperatures.

The research is of reference nature for peatlands occurring in this part of Europe, also because of the research technique, which used which provide continuous identification of ground

variability. The study considers electrical resistivity imaging results of the exemplary section of the Rospuda River Valley, which structure is original (no anthropogenic influence). The field study is located in the East-Central Europe, North-Eastern Poland (close to the border with Belarus) – Figure 1.

I suggest to add more information on the site. The Figure 1 is not described.

Figure 1: I suggest to improve the text with the indications on the features showed in the figure (i.e. dot line, vertical continue line, etc.).

• Thank you for suggestion. We have improved Figure 1 (now Figure 3).

Rospuda River Valley was formed as the peatland, which developed in the large land depression in the Holocene (deglaciation of a Würm-Vistuilan Phase – Figure 2). The valley of the Rospuda River is vast (in the region of analysis, it ranges from around 0,5 km to 1,5 km), which is related to the glacial history of this region. Nowadays, the meandering river is in the middle (south part of the research area) and close to the west bank (north part of the research area) of the valley mentioned above, which has around 2,5 km between geophysical prospection lines. The analyzed area is surrounded by a large forest complex, where the entire region is a Rospuda Valley Protected Landscape Area (which partly constitutes also the Professor Aleksander Sokołowski Rospuda Valley Reserve). Figure 2 presents geomorphology with the topographic situation, the location of archival geological cross-sections (Figure 3), and the location performed within these study geophysical prospection lines.

**2 Data description**

The text of the Data description chapter is missing several descriptions...the number of ERIs...how many roll-along were necessary,

• Thank you for pointing out this gap – we have a complete manuscript.

*For the first ERI profile performed using the roll-along technique, 7 stations (central unit positions - Terramter) were required. For the second, shorter ERI profile, 5 stations were necessary.*

The text on the Table 1 is more a table caption not a description.

• We have modified this manuscript fragment.

**Table 1: Typical electrical resistivity of materials (based on Tarnawski, 2020 and authors' experience).**

3 Experimental design, materials and method

This chapter should be merged with chapter 2.

We have merged this section with Chapter 2.

The figure 4 is not necessary, but it is important the sketch of the roll along distribution acquired data in order to check if there were some empty zone (zone without data) in order to check the final resistivity model.

**You are right. We have improved the manuscript in this field (Figure 4 has also been improved).**

4 Limitations: this chapter should be merged with chapter 5

**Thank you for suggestion – we have made adequate changes.**

5 Data Value: this chapter should be merged with chapter 4

**We have done so with reference to the previous comment**

In this chapter there are several considerations with only qualitative indications. I suggest to improve the discussion on the final results with more attention on the correlation between the geoelectrical results and the geological information. The final part is too generic and this part are only comments that did not improve the discussion.

**• Sorry for the gaps in the article. We have changed manuscript as you pointed out.**

The obtained electrical resistivity images revealed the geological structure and the position of the water table very well. On the electrical resistivity cross-sections it is possible to distinguish: (i) a zone of very high resistivity, which corresponds to sands and gravels with very low water content constituting the high ground (a few thousand  $\Omega m$ ; maximum ~4310  $\Omega m$ ); (ii) a zone of low resistivity (from ca. 75  $\Omega$ m to ca. 500  $\Omega$ m), occurring below the near-surface zone and which are most probably fully water saturated sands; (iii) a zone of even lower resistivity than Zone II (from ca. 40  $\Omega$ m to approx. 75  $\Omega$ m), which are water saturated silty sands (which may be overlain by clayey sands) - this zone is present at a relatively great depth of approx. 30 m; (iv) a zone of the lowest resistivity (10-40  $\Omega$ m), which is water-saturated peats of a thickness ranging from a few to several meters at the maximum (the highest thickness of peats is present near the Rospuda River). This interpretation corresponds to the geomorphology of the described area (detailed by Ber, 2007): the Rospuda River valley is a subglacial trough adjacent to sand plains in the west and east, within which numerous eskers and other forms of crevasses are present. This type of form was identified as an elevation in the first cross-section. The Rospuda River heads southeast, flowing approximately 2.5 km further into Lake Rospuda. Rospuda is a drainage river type, with groundwater recharge from the northwest and east. The groundwater table is in intense contact with the surface water of the Rospuda River. The influence of the water table level in the sand and gravel-type soils (Zone I) changes their electrical resistivity image (first cross-section), causing a difference in resistivity of about 1000  $\Omega$ m (a decrease from about 4000  $\Omega$ m to about 3000  $\Omega$ m). This description indicates a thick Quaternary groundwater aquifer (no impermeable soil to a depth of about 43 m) with no more significant surface isolation layers in the form of soils like clays or tills. In this context, peats constitute a separate hydrogeological layer characterized by lower hydraulic conductivity than the surrounding mineral soils but with excellent retention properties.

References: The number of cited papers is too short, I suggest to increase the number of the citation in order to improve the Introduction.

**Thank you for this suggestion. We have expanded the Introduction by doubling the number of references.**